# C19MOOC: A Remote Learning Reference Framework for Science and Technology Education

**Shahnawaz Saikat [1], Jaspaljeet Singh Dhillon [2,\*], Rosima Alias [3] and Mariam Aisha Fatima [4]**

[1]  College of Graduate Studies, Universiti Tenaga Nasional (UNITEN), Kajang 43000, Malaysia; saikatuniten13@gmail.com
[2]  College of Computing & Informatics, Universiti Tenaga Nasional (UNITEN), Kajang 43000, Malaysia
[3]  College of Energy Economics & Social Sciences, Universiti Tenaga Nasional (UNITEN), Kajang 43000, Malaysia; rosima@uniten.edu.my
[4]  Faculty of Health and Life Sciences, Management and Science University, Shah Alam 40100, Malaysia; mariam_aisha@msu.edu.my
\*  Correspondence: jaspaljeet@uniten.edu.my

**Abstract:** With the coronavirus disease 2019 (COVID-19) pandemic, education and learning have been compelled to go entirely online rather than using conventional offline media. As a result, remote learning adoption has exploded, neither planned nor anticipated. The challenges and benefits of such widespread adoption have gradually unfolded in front of our own eyes. Unlike other courses, science and technology (S&T) courses are more technical, involve practical lessons, and require careful planning for effective delivery via remote learning platforms. Thus, existing remote learning frameworks are too general and are not designed for S&T courses. In addition, the evolving requirements of learners' demand revision of prior frameworks to be relevant today. In this paper, we propose a remote learning reference framework called C19MOOC for S&T courses offered at higher learning institutions. The framework will provide the essential components to be considered in the development of remote learning systems for these courses. A focus group discussion was conducted to elicit learners' requirements and preferences for remote learning systems that offer S&T courses. The existing Massive Open Online Course (MOOC) framework was adapted to match learners' current needs and expectations. The MOOC framework's existing dimensions and components were redefined, and three new dimensions (*Engagement*, *Governance*, and *Self-determination*) and components were added. An expert review was administered to validate the C19MOOC framework. Based on our findings, it was determined that remote learning has excellent potential as an effective platform for education at higher learning institutions. Shortcomings that emerged during its immense use in the period of the COVID-19 pandemic can be addressed by leveraging the proposed framework. The C19MOOC framework will be useful for S&T education stakeholders, institutions, and system developers to identify suitable dimensions, components, and features to consider when designing remote learning systems for S&T education.

**Keywords:** remote learning; science and technology; focus group discussion; COVID-19

## 1. Introduction

Online learning, which takes place over the Internet, has increased in popularity to the point where it is now available in brick-and-mortar institutions that formerly only offered face-to-face instruction [1]. Remote learning is a subset of online learning that describes the process of learning across many locations via social and information exchanges via electronic devices, such as computers, cell phones, and wearable technologies [2–4]. Learners can grasp knowledge whenever and wherever they want, as long as they have a device linked to the Internet. Remote learning (also known as distance learning) allows learners to use instructional technology on their leisure devices [5]. It also enables learners to undertake educational tasks more simply by allowing them to remotely access learning

resources (lectures, courses, duties, quizzes, and exams) via electronic devices [6]. Recently, the application of remote learning to enhance teaching and learning among tertiary learners has piqued attention. While remote learning is rapidly becoming recognized as an effective way of improving education, it must continue to evolve to satisfy today's learners' changing requirements and needs [7]. Traditional teaching and learning methods are no longer appropriate or applicable. Remote learning is different from traditional ways of teaching and learning, as it is flexible, has an easy-to-follow learning path, is cost-effective, and prepares people for the digital future. As a matter of fact, remote learning has played a vital role in online education [8].

Education is crucial for everyone. Today, the delivery of education has changed tremendously. Most institutions are now geared toward implementing online education. Thus, remote learning is one of the most emerging prospects of online education. Through online education, all kinds of courses are offered and taught. Numerous courses are being delivered through remote learning platforms, including science and technology (S&T) courses. Technology courses refer to the use of scientific knowledge for practical goals, while science courses are the systematic study of the structure and behavior of the physical and natural world via observation and experiments. S&T courses require extensive lab work for practical knowledge purposes, and conducting these courses is quite different from other courses [9]. In a world where scientific knowledge is progressively increasing, technology is rapidly improving, and the influence of S&T can be seen in every aspect of our existence. We may comprehend the importance of scientific breakthroughs and inventions in contributing to the growth of countries and forming the foundation of scientific and technological improvements if we associate the world in which we live with a rich S&T class. Furthermore, a viewpoint such as this encourages science and education to grow significantly throughout time, thus pushing all nations to emphasize scientific development [10,11]. Therefore, S&T courses are deemed essential for education. Unlike other courses, S&T courses are more technical, involve practical lessons, and require careful planning for effective delivery via remote learning platforms. Practical facilities, course content, group work, and evaluation are vital; without these facilities, such courses will lose their educational value [12].

Since 2019, the entire world has drastically changed due to coronavirus disease 2019 (COVID-19), which originated in the People's Republic of China's Hubei Province and has spread across the world [13,14]. Due to the pandemic brought about by COVID-19, the traditional way of education could not proceed. A vast number of nations were affected by this virus. In late January 2020, the WHO Emergency Committee declared a global health emergency, citing a rise in COVID-19 confirmed cases worldwide [15]. COVID-19 spread across all continents, and every day in 2020 and even in 2021, the newest COVID-19 news flooded the global media [16]. The new variants of COVID-19 have been repeatedly wreaking havoc around the world [17,18]. This has drastically changed the learning activities of educational institutions. The educational institutions faced two completely different situations before and after the pandemic.

Before COVID-19, higher educational institutions mainly conducted their learning activities offline. On the contrary, institutions could not continue their education offline anymore, so they were forced to go online [19,20]. As a result, remote learning platforms have spiked due to the shift in education from offline to online. Therefore, the number of online learners has increased significantly, although the existing platforms are not capable of dealing with the immense number of users. Existing platforms are not designed or built to handle such a massive number of users, and they are too general for operational and technical functionalities. Thus, it was expected that educators and learners would experience several challenges in using these platforms to manage course materials, evaluations, and assessments [11,21]. For example, educators face many challenges while conducting online exams. Learners tend to cheat by looking for answers online or communicating with others during exams. Hence, it is challenging to maintain the quality of the assessment [22,23]. Most higher learning institutions do not have a platform to manage their

teaching and learning activities. Unfortunately, they had to rely on third-party platforms to proceed with online education, many of which were not designed for academic learning (e.g., assessments and evaluations) [24].

Existing platforms are mainly designed for conducting professional meetings, online calls, and casual group video calls. At the same time, video conferencing platforms download topped 62 million in March 2021 [25]. In online education, self-learning is essential, as learners do not have the opportunity to learn with others or get any sort of motivation from other learners. Existing platforms often lack features that encourage self-learning, e.g., enable learners to set personal goals [26]. It is hard to grab learners' attention throughout lectures in online learning. In addition, we cannot guarantee that the learner is present throughout the online class. Switching on cameras consumes a great deal of bandwidth and affects the connection. Hence, the attendance of learners in an online class can drop significantly [27]. Finally, lab facilities are essential for S&T courses; without these, learners do not get any hands-on experience. Every S&T course content is designed with practical work. However, existing platforms do not have any features for learners to engage in practical work. Therefore, the institutions could not provide remote lab facilities through these platforms. As a result, learners complete S&T courses without any practical experience [11,21,28].

We previously reviewed the benefits and challenges of existing M-learning (a form of remote learning) technologies used by learners and educators during the COVID-19 pandemic [29]. Five advantages and ten shortcomings were identified. The advantages were Technical, Operational, Financial, Resource, and Communication. In contrast, the shortcomings were Evaluation, Resource, Technical, Cope up, Psychological, Operational, Financial, Computer Literacy, Motivational, and Communication [29].

As a result, there is a pressing need for a remote learning reference framework that addresses the inadequacies of existing platforms and prioritizes the requirements of both learners and educators. The framework needs to be holistic, covering the current needs and expectations of both learners and educators. In this paper, we present the formulation of a remote learning reference framework for S&T education. A focus group discussion (FGD) was administered to elicit the needs and expectations of learners toward remote learning platforms designed for S&T courses. The established Massive Open Online Course (MOOC) framework was extended to meet the pressing needs of both learners and educators. An expert review was conducted to validate the framework.

The structure of this paper is as follows: Section 2 illustrates the existing frame-works and their applications. Section 3 presents the development process of the framework and the FGD used to gather the insightful needs and preferences of the learners toward remote learning. Section 4 presents the formulation of the proposed framework and its dimensions. Section 5 presents the validation of the framework. Sections 6 and 7 discuss the findings and limitations of this study, respectively. Section 8 concludes the paper and presents future work.

## 2. Related Work

Several frameworks might be employed to help in the development of remote learning. MOOC is one of the most famous online learning frameworks. MOOC is meant for online learning environments for huge groups of learners that are usually free, open, and adaptable. Online courses differ from traditional classes, as the former are large and open, allowing educators to engage with learners anywhere globally via an Internet connection [30]. Koutsakas et al. suggested a computer programmed hybrid MOOC framework combined with cMOOC and xMOOC for Greek secondary education. Educators and learners can share content that can be accessible to everyone, which will increase the quality of the resource and motivate the learners [31]. Don et al. proposed a conceptual business model for MOOC sustainability in higher education. The model is intended to serve as a guide for policymakers, practitioners, and researchers interested in ensuring the long-term viability of MOOCs in the hyperspeed era of innovation [32].

Marrhich et al. explained the eight dimensions of Khan Academy's MOOC framework for conducting online learning activities. The Khan framework's eight dimensions include administrative issues, such as leading organization and change, accreditation, and budgeting. Pedagogical issues include teaching to establish goals and objectives, content and approach design, methods, various learning strategies and activities, and technological issues, such as technology infrastructure, hardware, and software. Site design and usability are considered in the interface design. The assessment of learners and the learning environment is often referred to as evaluation. Maintenance of the learning environment is part of management. The ethical issue explores differences and legal concerns, such as privacy, plagiarism, and copyright, while resource support pertains to online support, such as educators, technicians, and other resources suggested in the learning environment [33].

Khan Academy's MOOC framework is renowned and widely used. In addition, many organizations such as Coursera, Udemy, edX, and many more use the MOOC framework as their base framework. Many researchers have also used this framework in their research. Therefore, the framework has incurred a massive amount of development. With time, the education sector is changing rapidly, and for this reason, there is a scope for amendment in the framework. This framework is highly structured and well designed for academic learning activities. It has been used to develop many problem-solving activities for S&T courses at various grade levels (from fundamental elementary concepts to higher-level topics) to help learners enhance their skills in science courses [33–36].

Additionally, MOOCs bring together individuals from all over the world and stimulate involvement between educators and learners in interacting with the general public. The popularity and acceptance rate for this framework is top-notch, which is why this framework is considered for the formulation of the proposed framework. The MOOC framework was selected and adapted as a base to develop the proposed remote learning reference framework called C19MOOC for S&T courses.

## 3. Research Work Process

The formulation of the C19MOOC framework undergoes a detailed and systematic step-by-step development process. Figure 1 shows the phases of the research process and the steps followed to meet the objectives of this study.

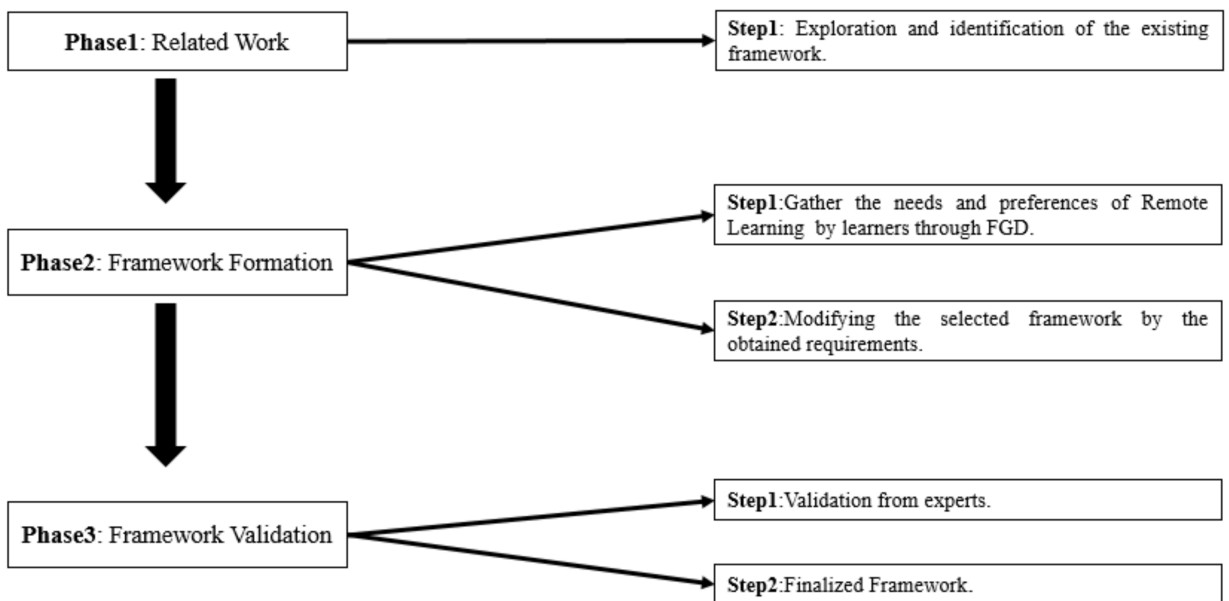

**Figure 1.** Workflow diagram of this study.

The research was conducted in three phases, and several activities were followed through in each stage. Phase 1–Related Work involved an exploration of existing remote

learning frameworks and identifying a suitable existing framework for the undertaken study. In Phase 2–Framework Formation, learners' needs and preferences toward remote learning systems catered to S&T courses were gathered through FGD. The selected framework (from Phase 1) was modified using the obtained requirements. Finally, in Phase 3–Framework Validation, the proposed framework was validated by experts.

An FGD was conducted with learners with a minimum of 1 year of experience in online education. The intention of conducting the FGD was to obtain insights into the challenges experienced by learners with remote learning during the pandemic and to comprehend their preferences and requirements for future remote learning systems designed for S&T courses. The FGD was preferred over other methods as it helps bring out the participants' spontaneous reactions and ideas on the topic of discussion [37]. FGD has allowed for in-depth talks but has also aided in the elicitation of subjective perspectives from participants through honest dialogs among peers [38,39]. The selected open-ended questions prompted during the FGD are listed in Table 1. These questions serve as a springboard for further debate and knowledge.

**Table 1.** Selected Questions of the FGD.

| No. | Questions |
|---|---|
| 1. | What are the most prominent challenges you faced with remote learning platforms for education? |
| 2. | What are your suggestions to overcome the challenges? |
| 3. | To address the current challenges, what features would you recommend to incorporate into remote learning platforms? |

FGD was conducted with 20 learners from higher learning institutions in Malaysia, and this was held in May 2021. All learners were pursuing postgraduate studies, and most had been doing online education for more than 1 year. The participants were of various age groups, genders, departments, and ethnicities. The age of the participants was not a factor in the selection process. The research took into account all of the participants' comments to obtain findings that were inclusive of the feedback and views supplied by participants from various backgrounds and countries. The FGD session lasted for 90 min and was facilitated by a moderator. At the beginning of the FGD session, the moderator provided an overview of remote learning and the study's intent. The participants were encouraged to express their thoughts, opinions, and suggestions. The questions were prompted during the session to prompt discussion. The participants presented themselves during the introduction session as a warm-up. The moderator's duty was to ask questions solely to stimulate and assist in the discussion session.

The participants were reminded to share their own thoughts and ideas throughout the discussion. Thematic analysis was used to analyze the findings obtained from the FGD analysis. This technique is noted to be a highly flexible, reliable, and effective method for exploring the views of various research participants, showing similarities and contrasts, and providing unexpected discoveries [40]. The FGD was audio-recorded and then transcribed. During the discussion, the participants' facial expressions were also monitored and considered. The accuracy of the notes taken during the FGD was compared with the audio recordings. The FGD transcripts were reviewed multiple times to emphasize the various challenges and suggestions for remote learning. One researcher retrieved the results independently and confirmed them with another research team member.

From the FGD, we were able to determine the problems learners had with online education and what they thought should be done. Thus, insightful information from the learners was gathered from the FGD. The gender, ethnicity, and department of the participants were all diverse, as indicated in Table 2.

**Table 2.** Characteristics of the FGD Participants.

| Category | Number and % of Participants |
|---|---|
| *Gender* | |
| Male | 17 (85%) |
| Female | 3 (15%) |
| *Ethnicity* | |
| Arabic | 10 (50%) |
| Malaysian | 6 (30%) |
| African | 3 (15%) |
| Bangladeshi | 1 (5%) |
| *Department* | |
| MIT | 9 (45%) |
| MEM | 6 (30%) |
| MBA | 5 (25%) |

The FGD included many participants since people from various cultures, places, ethnicities, and religions absorb information differently. Hence, various people had different opinions; thus, an insightful outcome was achieved. These diverse participants have pointed toward various scenarios for which the research paper holds useful information. However, the participants were occasionally silent and stopped sharing their opinions during the FGD. Therefore, the moderator had to prompt them with general related topics to continue the discussion, which helped them come back in rhythm. Although no ethical approval was needed, participants' consent was obtained before proceeding with the FGD session.

The outcome of the FGD was divided into 10 themes that emerged from the participants' responses: remote learning platform, connectivity, different time zones, motivational issues, learning content, distractions, technical issues, evaluation, practical and lab work, and standardized platform. These themes are explained in detail below.

*3.1. Remote Learning Platform*

The institution's instruction of medium was face-to-face before the outbreak of the COVID-19 pandemic. However, they were forced to change their medium of teaching from traditional face-to-face to online. Thus, learners were left to face problems coping with or adapting to the new teaching methods, as they were habituated to doing the classes offline. In fact, 8 of the 20 participants mentioned that they were facing adaptation problems with the new way of learning through remote learning. One participant shared the reason: "*I completed sixteen years of education through traditional face-face or physical classes, but since last one year I have to continue my education online which was very new for me. So, it is hard for me to adapt to the new way of learning*".

Moreover, institutions did not stick to a single platform, as they often changed their platforms such as Teams, Zoom, Google Meet, etc. Therefore, learners had to cope with new technology every time the institution changed its teaching platform preferences. Moreover, learners need to adapt to the new platform whenever institutions change their operational platform. For this reason, learners sometimes get confused. Another participant mentioned that "*Firstly, the online education was new for me and I am not a technical person so frequent change of platform by the institution, it's hard to cope up. Secondly, I struggled to learn the topic and the operational feature of the different platforms simultaneously*". These problems mainly occurred due to a lack of technical knowledge and the ability to adapt to the change in teaching delivery mode. Furthermore, the learners had to learn the operational understanding of the platform and the course content, which was different for the learners. Therefore, the learners' focus on their studies decreases.

*3.2. Connectivity*

Without the Internet, no online teaching is possible. For online education, the main component is the Internet connection. The connectivity issue has been identified as the most common challenge faced by all learners and educators. Classes could not be adequately conducted due to poor Internet connections. One participant indicated that he could not join the discussion early due to connectivity problems and missed about 15 min of the discussion. He also said that "*The Internet connection is not stable in his country Iraq. Furthermore, in Iraq, the connectivity differed from town to town. Especially, the big cities had good coverage rather than the small cities*". The connection to the Internet depends on the geographical location. Over the world, the Internet connection is not the same. Internet connections can vary in terms of bandwidth, stability, and expenses among developed, developing, and underdeveloped countries. The infrastructure of the Internet is poor in small towns and in the villages of the countries. Even some of the country's towns did not have any facilities for it. Another participant shared his view that "*the Internet connection is also dependent on the ISP (Internet Service Provider). Sometimes I had trouble with the connection, but my friend did not feel any issue as we used different vendors' Internet connections although we lived in the same city*". In addition, the connection was distinct due to the different Internet service providers, even in the same city in the country.

Furthermore, the Internet also depended on the country's weather and electricity supply. Some towns or villages in underdeveloped and even developing countries had huge load shedding problems that caused connectivity problems. A Nigerian participant shared, that "*In my country, there is a lot of problem with the supply of electricity. So, due to interruption of power, the Internet connectivity is inferior. It's the same issue in nearly every African country*".

*3.3. Different Time Zones*

With online education, the most common problem that international learners experience is varying time zones. Institutions often conduct their classes according to the timetable; however, international learners were taking the course in their respective countries. As a result, the time zones of the learners and the institutions can be different. It often leaves learners to do live classes, online presentations, and evaluations at inappropriate times, such as midnight or early in the morning. For this reason, learners are not able to fully concentrate on their studies, which affects their learning outcomes and assessment results. One participant living in Iraq and continuing her studies in Malaysia shared: "*I am pursuing a master's degree in Malaysia. My classes were often held in the afternoons around 6 p.m. However, I am now residing in Iraq, so I am required to attend lessons according to Malaysian time, which is quite strange for me because the classes begin after 1 a.m.*".

*3.4. Motivational Issues*

On campus, students often participate in extracurricular activities that play a crucial role in strengthening their social bonds with classmates, which is critical for their academic success. Learners often become motivated by their classmates through constant interactions with each other. On the contrary, remote learning does not provide an option for learners to engage in cocurricular activities that foster interactions between learners. Therefore, traditional face-to-face learning is valued more in this case. Participation in sports and cultural events of the institution helps broaden the "learner's motivation to study". One participant mentioned that "*When I see my classmates succeeding and working hard, it motivates me to study hard and pay more attention in my lessons*".

*3.5. Learning Content*

In education, learning resources are essential for every learner. Unsuitable learning materials and delivery modes can affect the learning outcomes of learners. A participant shared her view that "*Sometimes online content is not accessible due to server maintenance or crash, but offline is always accessible if I have the content*". In face-to-face teaching, the content

can be made available offline, so there is no dependency. Learners can go through them whenever they want. Moreover, in online teaching, the classes' content must be different from that of traditional face-to-face classes. However, most of the educators used the same class materials when, in fact, online class materials must be different and more dynamic than those used in offline classes. The class should be more discussion-based, and the content should not be too theoretical. One Bangladeshi participant stated that "*the content of the online classes was the same as offline classes, so it is very boring to do the long class I lose my concentration. I think the online content must be more dynamic and interactive so that learners do not lose the attention*". Even from the institutions, there was no monitoring of the course content by which the educators conducted the online classes. Another participant indicated, "*We fill out a recommendation form at the end of the semester to suggest ways to make the course more successful, but nothing changes in the next semester*". For this reason, learners were dealing with the problem of doing online classes, and the effectiveness of the education was declining.

### 3.6. Distractions

Online teaching allows flexibility and mobility, as learners can now attend online classes from anywhere. However, a friendly environment is not suitable for learning, where learners can easily be distracted by family members, friends, and housemates. One participant shared her thoughts: "*Doing classes from home is very disturbing because of my children. They do not let me to attend the classes properly*". It is a challenge for learners to concentrate during class. For a friendly atmosphere, psychology is different from a face-to-face environment. In offline classes, there is a minimum distraction. Learners can focus on the class by having their educators in front of them.

In contrast, the educator has less control over the learners and their activities during online classes. Learners can attend the course and do other things at the same time. Another participant mentioned, "*We do not use our phones in class, but we use them for messaging or receiving calls at home when taking lessons. As a result, there is a significant difference in attention levels*".

### 3.7. Technical Issues

Several technical and operational difficulties were noted with the platforms used for online instruction. In 2020, Zoom was hacked for a short time [41]. As a result, Zoom users were compelled to switch to other platforms. A participle mentioned that "*We used Zoom for our online lectures at first, but the institution switched to Microsoft Teams when there were issues with the platform. Because the two platforms operated differently, I had to deal with each one independently*". The platform's free version was severely limited in functionality, while the licensing or subscription version was too expensive for several institutions. The educators could not manage the learners' microphones during the classes, so if more than one student spoke or anyone made a noise, the entire class was disrupted. Moreover, while taking lessons, the platform would occasionally crash, or some functions would not operate properly. There was some problem, even with the microphone. After restarting the platform, the problem often disappears. As a result, the learner's attention was broken, making it difficult to focus on the class lesson. Another participant shared his perspective: "*I had many problems with the sound quality of the instructors' microphones and even noise from the learners. The video of the recorded classes is sometimes inaccessible for download or watching due to server crash or maintenance*".

### 3.8. Evaluation

All the participants considered evaluation to be another common problem. One of the participants suggested that "*I need to do some extra work while doing the online assessment like paraphrasing and checking for plagiarism to make answer unique, which takes time away from the exam. As a result, we cannot just concentrate on answering questions. As a result, I did not get a good grade in the online exam*". Educators must also make an extra effort to ensure

that students do not copy and paste answers from the Internet, class notes, or even from other students. Another participant shared his view: "*The average mark obtained through online assessment is better than that obtained through offline evaluation, although higher grades are difficult to obtain. Learners have access to external sources to provide answers to each test question. When it came to giving grades, instructors in the offline edition were more liberal than in the online version*".

*3.9. Practical and Lab Work*

Learners studying S&T encountered significant challenges in completing practical and laboratory work. In the virtual world, providing laboratory facilities is difficult. A hands-on laboratory setting is required for students studying medical and engineering courses (e.g., civil and mechanical). Online education has negatively impacted courses that rely on hands-on lab work since educators are unable to provide the necessary lab instruction. One participant shared his opinion: "*The majority of undergraduate students, especially in Civil Engineering, Mechanical Engineering, or Medicinal field, all the fields need laboratory work, which will encounter many obstacles during practical sessions, as conducting lab sessions for these programs* via *online education is nearly impossible. On the other hand, Computer Science and Electrical Engineering labs can be conducted online as many simulations software are available*". Educators of computer science and electrical engineering can use simulated software, such as MATLAB, ETAP, Proteus, Cisco packet tracer, Visual Studio, NetBeans, Atom, and many more. Therefore, the learners faced fewer problems while studying these courses and fewer difficulties with lab work. As a service, this software may be connected to a standardized platform, allowing learners to acquire practical knowledge.

*3.10. Standardized Platform*

Five of the participants mentioned that there should be one common platform where learners can do their classes and assessments. Due to the plethora of platforms employed by instructors, it was challenging for students to continue their education online. One participant mentioned that "*I have had to use many different platforms. Most of the instructors used Zoom and later Microsoft Teams in the first semester. Sometimes, different instructors use different platforms, which is troubling because it takes time to get used to a platform*".

Several participants suggested that there should be a tutorial video on the settings of the platforms used. In fact, one of the participants stated that "*an operating manual of the platform will be beneficial for students because I felt a lot of difficulties when I used it for the first time*".

Some additional features or tools can be incorporated to attract learners' attention to online classes. For example, one participant mentioned, "*In Data Analytics classes, Kahoot video game was used to make the class interactive, which draws my attention toward classes than other classes*". Another participant shared that "*in last semester we used Padlet where I can see the other classmate's view. So, it takes less time to understand the topic*". For this reason, the majority of the participants suggested a single standardized platform where they could do all courses and the assessment. This will lessen the pressure learners often experience; thus, they can be more focused on their studies.

The learners' most significant need and preference was the use of one standardized platform instead of using different platforms in order to minimize the shortcomings faced by the learners while doing S&T courses at higher institutions. Most learners preferred remote learning over offline learning in higher institutions, provided that the platforms' challenges were addressed or minimized.

**4. The Proposed C19MOOC Remote Learning Reference Framework**

As mentioned in Section 2, Khan Academy's MOOC framework is well known and is often used in creating other digital education frameworks. In fact, popular learning platforms, such as Coursera, Udemy, and edX, are built on the MOOC framework. In recent years, especially during the COVID-19 pandemic, the education industry has reached a

turning point at which major amendments are gradually coming into place. Hence, there is room for improvement in the MOOC framework to reflect the ongoing changes. The MOOC framework was found to be suitable and was adapted to meet the objective of this study in aiding the development of a remote learning reference framework for S&T courses. The proposed C19MOOC framework is the essential supporting structure of our solution to address the current shortcomings of remote learning. A modification was made based on the results of the FGD study presented in Section 3.

The C19MOOC framework, as shown in Figure 2, has eight dimensions, and each dimension has two or three components, which are explained below:

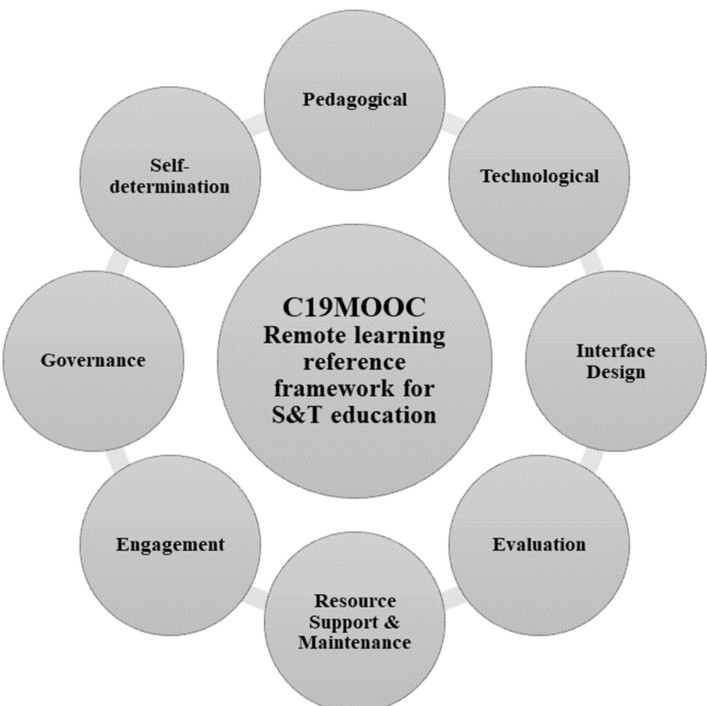

**Figure 2.** Remote learning reference framework for S&T education (C19MOOC).

*4.1. Pedagogical*

Components: Analysis, Strategy, and Diversity

Teaching and learning strategies are included in the pedagogical part of remote learning. Goals/objectives, content, design approach, organization, techniques, tactics, and the medium of remote learning environments are all covered in this dimension. Presentation, demonstration, drill and practice, tutorials, games, storytelling, simulations, role-playing, discussion, interaction, modeling, facilitation, collaboration, debate, field trips, apprenticeship, case studies, generative development, and motivation are just a few examples of remote learning methods and strategies.

*4.2. Technological*

Components: Development and Sustainability, Functionality, and Integration of Third-Party Software, and Accessibility

This technical infrastructure must be routinely developed and updated regularly. Although not restricted to these events, the technology infrastructure includes the Learning Management System, Online Class Section System, and authoring tools. This element denotes the technological infrastructures and assets that make up the backbone of a remote learning entity. The technical infrastructure includes equipment, processes, and applications whose effectiveness may be measured in terms of availability and dependability, appropriate functionality, usability, and integration with the existing infrastructure.

Validation of data collection, storage, and retrieval should be specified in a written strategy to assure dependability and integrity. The foundation of remote learning should be reviewed regularly to adapt to new problems. For an effective remote learning platform, the platform must be capable of adding new functionalities or features. Learners' comments should be collected regularly, considering the remote learning infrastructure's simplicity of use, efficiency, robustness, and reliability.

The remote learning infrastructure is known to be a complicated system in which highly interconnected technical aspects support many connections and interactions. In the case of a system failure, there should be a reasonable response time. The technical platform's functionality should be evaluated on a frequent basis. In remote learning contexts, sensitive information should be kept private and secure. Some features of artificial intelligence can also be considered. For example, if someone forgets to turn off the microphone after speaking, a message will pop up saying "turned off the microphone". A CAPTCHA (Completely Automated Public Turing Test to Tell Computers and Humans Apart) can be shown every 15 min or more during online classes to make sure learners are there. Software from third parties, such as MATLAB, ETAP, Proteus, Cisco packet tracer, Visual Studio, NetBeans, and Atom, can be used to create virtual labs; thus, these should be added to the remote learning platform. Furthermore, quizzes in class must be organized utilizing gaming applications such as Kahoot and Padlet to capture and motivate learners.

*4.3. Interface Design*

Components: Page, Content, Site Design, Accessibility, and Usability

Usually, remote learning platforms are static and not user friendly, and the interface is not lucrative enough to grab learners' attention. Therefore, a user-friendly environment, a suitable prospective path, and an axiomatic remote learning platform should be provided by developing a user-friendly remote learning environment to execute learning activities efficiently. The cognitive burden may be reduced by using the right color and pattern. The graphical interface needs to be more dynamic, and the use of scripting language will help the platform to be more responsive. Moreover, giving the user visual help through the appropriate use of graphics, text, photos, music, video, animation, and other media can provide search choices that are convenient to use. Learners can navigate a fully linked network with standardized navigation with minimal effort. The information should be put in a certain order or place to make it easier to find.

*4.4. Evaluation*

Components: Assessment of Learners, Evaluation of Content and Resource, and Evaluation of Educators

Evaluation has been identified as a crucial part of learning. Any educational institution must guarantee that assessment proposals are open and secure. Learners often require evaluation, which must be carried out flawlessly. Remote learning can improve the evaluation by additional features, e.g., freezing the screen (locks up the particular window) during assessments and turning the video camera on to reduce cheating. Continuous video recording without Internet connection and then uploading the video recording automatically should be done. An Internet connection is needed to start the assessment, but it should work as offline software once the evaluation commences to prevent cheating. The technique will remain the same even if the connection is lost. The answer sheet and the lost session should be submitted to the system right away. Then, educators should be able to assess them properly, similar to face-to-face assessment. Furthermore, online education's learning materials and resources should be evaluated regularly to maintain educational quality. After completing a course, there should be an online survey in which learners' feedback is considered, and the course should be updated accordingly in the future.

### 4.5. Resource Support and Maintenance

Components: Availability, Online Support Assistance, and Content Management

The resource is the most essential part of education, as nothing can be accomplished without it. Remote learning resources should be more dynamic and engaging than traditional face-to-face resources. Institutions must guarantee that help is available. In relation to this, educators can deliver taped class lectures (if there is a geographical variation), videos, a live component, or perhaps both. A guidebook or demonstration video on operating and using the system can be distributed to students and educators. Continuous maintenance is required to ensure the platform's and resource's availability. The mirror server and any cast routing may be used for this. There should also be a dedicated support team (preferably available 24 h a day) that will ensure that all geographically scattered students have access to the educational materials. As a result, they prefer to receive some type of aid, such as technical assistance, counseling, and other types of assistance. In addition, maintaining accurate and consistent learning resources requires regular monitoring. Learning resources should be appropriately examined and updated with regard to internal assessment and improvement.

### 4.6. Engagement

Components: Community of Learners and Discussion between Learners and Educators

Interaction (between learners and educators, between learners and learning environments) is at the heart of learning. It can foster discourse between and among all learners in a remote learning environment. More elements are needed in remote learning settings than in campus-based education to achieve better engagement. Thus, there should be an interactive discussion community where learners can interact with themselves, share their views, discuss class materials, and build friendly relationships. There should also be an option to interact with educators about their problems and discuss them. Constructive feedback, such as reinforcing learning, being genuine, correcting mistakes, and providing information in context, should be given on time. A range of communication modalities should be used to offer in-depth and relevant feedback.

### 4.7. Governance

Components: Establish Rules and Regulations, Surveillance, and Educators' Governance

The Education Ministry should establish guidelines and policies for institutions to follow while conducting online education. In addition, institutions should also provide rules and regulations for educators and learners. Learning objectives should be explicitly linked across learning and evaluation activities. A concise and comprehensive course description, as well as a syllabus, should be provided. The educational framework or methodologies and the requisite level of participation should be explicitly stated. To preserve the quality of education, everyone must adhere to rules and regulations. As online education requires more attention than conventional face-to-face teaching, educators should closely monitor each student and offer them tailored guidance.

### 4.8. Self-Determination

Components: Competence, Connection and Relatedness, and Autonomy

Self-determination is identified as a key notion related to each individual's ability to make decisions and govern their own lives. This capacity is crucial to one's mental health and well-being. People who have self-determination have a sense of control over their decisions and lives. It also influences motivation; individuals are more driven to act if they believe their actions will impact the results. Self-determination has been used in several fields, including education. Having a high level of self-determination may lead to success in various areas. In online education, self-learning is critical, as the learning material is accessible, and learners are expected to study it independently. Thus, self-determination plays an important role. Additional features can enable learners to set goals, such as project management tools and intrinsic rewards, which will motivate them to achieve something

to avoid mental pressure. Learners must feel in charge of their actions and objectives. This notion of taking direct action that will result in genuine change contributes significantly to the learner's feelings of self-determination. Learners must master assignments and acquire a variety of abilities. Leaners are more inclined to take activities to help them reach their goals if they believe they have the necessary abilities.

*Engagement*, *Governance*, and *Self-determination* are the three new dimensions added to the existing framework. The remaining four were inherited and modified according to the requirements.

Table 3 summarizes the dimensions with their definitions, components, and additional features identified from the FGD study to address the shortcomings of remote learning.

**Table 3.** Dimensions, Definitions, Components, and Additional Features of C19MOOC Framework.

| Dimensions | Definition | Components | Additional Features (to Address the Identified Shortcomings of Remote Learning) |
|---|---|---|---|
| Pedagogical | Teaching and learning are referred to as the pedagogical dimension of remote learning. This dimension includes content analysis, audience analysis, goal analysis, media analysis, design approach, organization, and remote learning environment methodologies and tactics. | • Analysis<br>• Strategy<br>• Diversity | N/A |
| Technological | The remote learning framework's technological dimension investigates concerns of technology infrastructure in remote learning settings. Infrastructure planning, hardware, and software are all included. | • Development and Sustainability<br>• Functionality and Integration of Third-Party Software<br>• Accessibility | Integration of software such as MATLAB, ETAP, Proteus, Cisco packet tracer, Visual Studio, NetBeans, and Atom will provide the lab facilities and capture learners' attention in the class. Artificial intelligence features, e.g., the mic gets muted if someone is not talking. Use CAPTCHA to verify the attendance of learners. |
| Interface Design | The entire appearance and feel of remote learning applications are referred to as interface design. Page and site design, content design, navigation, and usability testing are all part of the interface design dimension. | • Page, Content, and Site Design<br>• Accessibility and Usability | The interface should be more dynamic and responsive by enabling an enhanced learning user experience. |
| Evaluation | The assessment of learners as well as the evaluation of the teaching and learning environment are both included in the remote learning evaluation. | • Assessment of Learners<br>• Evaluation of Content and Resource<br>• Evaluation of Educators | Freezing the screen (locks up the particular window) during assessments and turning the video camera on to reduce cheating. Continuous video recording without the connection and then uploading the video recording automatically. An Internet connection is needed to start the assessment, but it should work as offline software once the evaluation commences. |

**Table 3.** *Cont.*

| Dimensions | Definition | Components | Additional Features (to Address the Identified Shortcomings of Remote Learning) |
|---|---|---|---|
| Resource Support & Maintenance | The resource support and maintenance component addresses the online assistance and resources necessary to build effective learning environments and the maintenance of learning environments and information dissemination. | • Availability<br>• Online Support Assistance<br>• Content Management | Mirror server technique and anycast routing be implemented, and there should be a 24-h support team to provide help when needed. |
| Engagement | The contact between learners and educators will be ensured via *Engagement*. It will boost communication, motivation, and interaction among learners. | • Community of learners<br>• Discussion between learners and educators | |
| Governance | The *Governance* will be maintained via hierarchy, which means that education ministry is at the top and learning institutions, educators, and learners are below sequentially. The hierarchy will establish rules and regulations and will ensure that everyone follows the norms. | • Establish rules and regulations<br>• Surveillance<br>• Educators' governance | |
| Self-determination | The self-study, set up goals, time management for individual and also the individual completion and progress is included in the *Self-determination* dimension. | • Competence<br>• Connection and Relatedness<br>• Autonomy | |

## 5. Validation of the C19MOOC Framework

Expert reviews were conducted in December 2021 to validate the proposed C19MOOC framework. The Delphi approach is used with experts in several rounds, either as a series of successive surveys or interviews, to determine the most credible consensus of viewpoints [42,43]. Delphi is well known for its accuracy, and it has been frequently used in the sectors of computers, information systems, and education [44,45]. As a result, this method was utilized to test the framework with experts and to further facilitate interactive and organized dialog with specialists to gauge their perspectives, as well as their preferences and suggestions to improve the proposed framework. Five experts were involved in the validation of the C19MOOC framework. The experts were of S&T background; all were educators with a PhD, and they were recruited randomly from various educational institutions.

The expert reviews were conducted using a qualitative interview preceded by the experts presenting their credentials. Each interview lasted around 60 min. At the beginning of the session, the interviewer presented the proposed framework, its dimensions, components, and additional features. The validation process involved the experts going through each dimension, definition, and component of the C19MOOC framework while ticking off any degree of agreement or disagreement on assertions. They were also asked to give feedback on the validation form, and they expressed their opinions during the interviews. The interview was audio-recorded and then transcribed. The frequency count analysis approach was used to analyze the expert review data. By counting, the frequency count analysis approach may provide more valid and appealing arguments [46]. Furthermore,

this approach was used to tally the number of times the degree of agreement or disagreement was expressed in the research [47]. This expert review research aims to determine the degree of agreement or disagreement on assertions regarding the framework, namely the contributing aspects or dimensions. The questions were intended to assess the degree of agreement or disagreement with claims to validate the framework. As a result, the frequency and percentage of the statements' degree of agreement or disagreement are tallied and computed using the frequency count analysis method [48].

Strongly disagree (D3), moderately disagree (D2), slightly disagree (D1), neutral (N), slightly agree (A1), moderately agree (A2), and strongly agree (A3) are the seven degrees of agreement utilized in the questions [46,47]. Combining the A1, A2, and A3 replies yields a percentage of agreement (% Agree). In Table 4, the frequency count analysis is presented, where the five experts' responses are noted as E1, E2, E3, and so on, according to their degree of agreement.

**Table 4.** Frequency Count Analysis of the C19MOOC Framework.

| Dimensions | Components | D3 | D2 | D1 | N | A1 | A2 | A3 | % Agree | Comments of the Experts |
|---|---|---|---|---|---|---|---|---|---|---|
| Pedagogical | • Analysis<br>• Strategy<br>• Diversity | | | | E3 | | | E1, E2, E4, E5 | 100 | Pedagogical is an important factor as the students are of different backgrounds. The approach must be properly designed to suit the students [E5]. |
| Technological | • Development and Sustainability<br>• Functionality and Integration of Third-Party Software<br>• Accessibility | | | | | | E2, E5 | E1, E3, E4 | 100 | Technology provides students with easy-to-access information, accelerated learning, and fun opportunities to practice what they learn [E5]. |
| Interface Design | • Page, Content, and Site Design<br>• Accessibility and Usability | | | | | | E1, E3, E5 | E2, E4 | 100 | It creates fewer problems, increases user involvement, perfects functionality and creates a strong link between the students and remote learning or the content [E5]. |
| Evaluation | • Assessment of Learners<br>• Evaluation of Content and Resource<br>• Evaluation of Educators | | | | | | | E1, E2, E3, E4, E5 | 100 | It is important to understand how students learn and where they need help [E2]. Evaluation is an important component of the teaching-learning process. It helps lecturers and learners to improve teaching and learning [E5]. |
| Resource Support & Maintenance | • Availability<br>• Online Support Assistance<br>• Content Management | | | | | | E2, E5 | E1, E3, E4 | 100 | |
| Engagement | • Community of Learners<br>• Discussion between learners and educators | | | | | | | E1, E2, E3, E4, E5 | 100 | Engaging students in the learning process will increase their attention and focus, motivate them to practice higher-level critical thinking skills, and promote meaningful learning experiences [E5]. |

**Table 4.** *Cont.*

| Dimensions | Components | D3 | D2 | D1 | N | A1 | A2 | A3 | % Agree | Comments of the Experts |
|---|---|---|---|---|---|---|---|---|---|---|
| Governance | • Established rules and Regulations<br>• Surveillance<br>• Educators' Governance | | | | | E2, E3, E5 | E1 | E4 | 100 | Flexibility may help design a more effective learning framework. Keep in mind how to extend the framework when new techniques are discovered [E2]. This can be controlled by classifying users. But definitely need to have a set of rules [E4]. |
| Self-determination | • Competence<br>• Connection and Relatedness<br>• Autonomy | | | | | E2, E5 | | E1, E3, E4 | 100 | *Self-determination* can be extended with Certification. When students participate in a remote learning, it could be a professional course. Just like the courses offered by Udacity, Coursera, etc. Students can share those certificates in their resumes [E4]. It will help the students learn how to participate more actively in educational decision-making by assisting them in becoming familiar with the educational planning process [E5]. |

Among the three new dimensions, *Engagement* is the most agreed dimension. All the experts strongly agreed with the *Engagement* and its components. For example, one of the experts shared, "*engaging students in the learning process will increase their attention and focus, motivate them to practice higher-level critical thinking skills, and promote meaningful learning experiences*".

The experts warmly accepted the Self-determination dimension of the framework. The experts commented that "It will help the students learn how to participate more actively in educational decision-making by helping them become familiar with the educational planning process". In remote learning, self-learning is one of the most challenging parts of ensuring self-determination; the problem can be overcome, according to the expert review. Another expert suggested that "*Self-determination can be enhanced if certifications are provided upon completion of a course. When students participate in remote learning, it could be a professional course. Just like the courses offered by Udacity, Coursera, etc. Students can share those certificates in their resumes*". This suggestion was included in the competence component.

All the experts agreed with the proposed C19MOOC framework's dimensions and components. All preferences and suggestions were considered carefully and modified according to the framework. However, the framework had no significant changes, as all the experts agreed with the proposed one. One of the experts commented that "*the Pedagogical and Governance dimension was not clear how they are relevant to remote learning*" [E3]. According to the suggestion, the framework was modified, and the *Pedagogical and Governance* were defined and described more clearly. Thus, C19MOOC has the potential to be a standardized framework for developing S&T courses on remote learning platforms.

## 6. Discussion

The pandemic brought about by COVID-19 has had a tremendous impact on education and other aspects of life throughout the world. First, educational institutions have decided to shift from face-to-face education to remote learning. These adjustments were made

to protect public health. These institutions use a variety of platforms to deliver online programs. In terms of functionality and operation, the remote learning platforms were noted to vary. As a result, the learners' attention was drawn to their challenges in adapting to diverse platforms. The majority of the educators utilized a preferred platform for conducting classes and evaluating learners. For this, learners had to use several platforms for various courses. However, in some cases, the platforms utilized for evaluation are entirely new to the learners compared to what they had used before. Therefore, it made the evaluation considerably more difficult for them. In that situation, the learners felt confused about whether they would learn about the platform's operation to proceed with the assessment.

For this reason, most participants suggested a single standardized platform where they could do all courses and the assessment. This will decrease extra pressure on the learners, and they can be more focused on their studies. Moreover, lab work facilities are also missing from the existing framework. To address the absence of virtual laboratories, the proposed C19MOOC has included the *Functionality and Third-Party Software Integration* components in the *Technology dimension*. Furthermore, the lack of communication between learners and educators can be minimized by including the *Engagement* dimension in the framework. As a result, both stakeholder education ministries and educational institutions have failed to develop policies for remote learning. Education is losing value for the time being.

The proposed framework has been identified as playing a vital role in developing a standardized remote learning platform for higher institutions. This study addresses the shortcomings gathered from the FGD by revising and extending the existing MOOC framework of Khan Academy to formulate the C19MOOC framework (see Table 5). The proposed framework was validated by experts and amended according to their suggestions. Three new dimensions, namely, *Engagement*, *Governance*, and *Self-determination*, were added to meet the requirements and expectations of learners toward remote learning focused on S&T education.

**Table 5.** Formulation of the C19MOOC Framework.

| Dimensions | Adapted from the MOOC Framework | Remarks |
|---|---|---|
| Pedagogical | Pedagogical + Institutional | Directly inherited |
| Technological | Technological | Inherited and modified |
| Interface Design | Interface Design | |
| Evaluation | Evaluation + Ethical | |
| Resource Support & Maintenance | Resource Support + Management | |
| Engagement | | New |
| Governance | | |
| Self-determination | | |

In this present study, we aimed to establish a remote learning reference framework to address the current needs of educators and learners for S&T education. The scope includes examining the deficiencies of current remote learning practices [29], eliciting the needs of learners for remote learning, developing the C19MOOC remote learning reference framework, and validating the framework with experts. We have also created a checklist (see Table 6) based on the C19MOOC framework to assist application developers and education stakeholders in developing remote learning systems for S&T education. Specific graphical structures of possible remote learning systems are not provided, thereby enabling application developers or education stakeholders to design systems for S&T according to their requirements and preferences, as long as the framework and checklist-suggested dimensions and components are presented.

**Table 6.** C19MOOC remote learning reference framework checklist.

| Dimensions | Definition | Components | ✓ |
|---|---|---|---|
| **Pedagogical** | The pedagogical part of remote learning is defined as teaching and learning. This dimension encompasses content analysis, audience analysis, goal analysis, media analysis, design approach, organization, and techniques and strategies for remote learning environments. | **Analysis** The system enables the analysis of the learning content, target audience, and goal. | |
| | | **Strategy** The system enables the development of organizational strategies and designs. | |
| | | **Diversity** The system allows learners of diverse cultures and locations to participate spontaneously. | |
| **Technological** | The technological part of the remote learning framework explores questions about technology infrastructure in remote learning environments. Infrastructure design, hardware, and software are all part of the package. | **Development and Sustainability** The system is sustainable in the long run and flexible enough to meet the changing needs of learners. | |
| | | **Functionality and Integration of Third-party Software** The system incorporates AI features that make it more valuable (e.g., it reduces distractions during lectures) and permits the integration of third-party software to solve current deficiencies (e.g., the lack of virtual lab facilities). | |
| | | **Accessibility** The system is accessible to authorized learners. | |
| **Interface Design** | Interface design refers to the look and feel of remote learning systems as a whole. The interface design dimension encompasses page and site design, content design, navigation, and usability testing. | **Page, Content, and Site Design** The system's dynamic page design and responsive content enable learners to focus on the learning content. | |
| | | **Accessibility and Usability** The system's user interface is intuitive, and only authorized users can access the learning materials. | |
| **Evaluation** | The remote learning evaluation includes both the assessment of learners and the evaluation of the teaching and learning environment. | **Assessment of Learners** The system provides a secure and suitable environment for examination in which students cannot cheat. | |
| | | **Evaluation of Content and Resource** The system's AI features help in evaluating the learning content and resources that are updated by educators. | |
| | | **Evaluation of Educators** The system evaluates educators based on student feedback after each course is completed. | |
| **Resource Support & Maintenance** | The online assistance and resources required to develop successful learning environments, as well as the maintenance of learning environments and information distribution, are addressed in the resource support and maintenance component. | **Availability** The system allows authorized learners to access the learning content at any time and anywhere. | |
| | | **Online Support Assistance** The system features a Chatbot and/or support team to assist learners with system-related issues. | |
| | | **Content Management** The system manages online learning resources based on the needs of the learners. | |
| **Engagement** | Engagement will guarantee that learners and educators are in communication. It will improve student communication, motivation, and interaction. | **Community of Learners** The system enables learners to connect and converse with one another, which helps enhance their communication abilities, motivation, and interaction. | |
| | | **Discussion between Learners and Educators** The system facilitates discussions between learners and educators on lessons learned. | |

**Table 6.** *Cont.*

| Dimensions | Definition | Components | ✓ |
|---|---|---|---|
| **Governance** | The governance structure will be based on a hierarchy, with the education ministry at the top and learning institutions, educators, and learners following in order. The hierarchy will set rules and regulations, as well as ensure that everyone adheres to them. | **Establish Rules and Regulations**<br>The system enables a higher authority to enforce laws and facilitates monitoring and reporting to the respective authorities. | |
| | | **Surveillance**<br>The system enables the relevant authorities to monitor and enforce compliance with the law. | |
| | | **Educators' Governance**<br>The system monitors educators' governance of learners, which is later reflected in educators' performance. | |
| **Self-Determination** | The Self-determination component includes self-study, goal setting, time management for individuals, as well as individual completion and progress. | **Competence**<br>The learning materials presented in the system motivate learners to engage in self-learning. | |
| | | **Connection and Relatedness**<br>The system allows learners to assess themselves and get immediate feedback on their answers after taking an exam. | |
| | | **Autonomy**<br>The system enables students to learn at their own pace and according to their preferred learning style. | |

## 7. Limitations

Although the developed framework is deemed adequate and considers both educators' and learners' perspectives, the study still has certain limitations. To begin with, there was only one FGD conducted among the learners. The learners who participated in the FGD were postgraduate students; hence, they were not representatives of lower-level learners. However, the needs and preferences obtained through FGD were sufficiently insightful to conduct this study. Due to COVID-19, it was challenging to look for experts to review the framework. Only five specialists were involved in the validation process. In spite of the framework being tested by just a few individuals, it was sufficient since we performed separate sessions with each of the experts.

## 8. Conclusions

Despite the benefits of remote learning for educators and learners, several challenges remain to be addressed. During the COVID-19 outbreak, learning activities shifted from offline to online overnight. Many learners, especially those enrolled in S&T programs, had a difficult time adjusting to such rapid changes, and thus endured the trouble of online learning. During this period, the immense use of remote learning brought many insights that helped us understand how remote learning can be further improved in meeting learners' and educators' evolving needs and preferences. Thus, in this study, we proposed a remote learning framework called C19MOOC to aid developers in designing remote learning systems catered to in S&T education. The framework was gradually presented, starting from understanding its needs, comprehending the challenges faced by educators and learners, reviewing existing solutions, eliciting the needs and expectations of the learners, conceptualizing the C19MOOC framework, and validating the framework with experts.

To overcome the shortcomings of existing frameworks, it was essential to determine the needs and preferences of learners who had experienced online education. The proposed framework was extended from Khan Academy's MOOC framework according to the needs and preferences obtained from the learners through an FGD. The framework was then validated by experts, which increased the framework's acceptance. The framework will be helpful for application developers in establishing remote learning platforms for higher education institutions. In addition, the framework will aid education stakeholders and institutions in determining the impact of using remote learning as an exclusive mode

of instruction. Despite its popularity and need during the pandemic, the use of remote learning will continue to increase since everyone is habituated to it, and it is envisioned to expand rapidly. This is why there is a huge scope for this platform, and the C19MOOC framework is just the beginning phase.

*Engagement*, *Governance*, and *Self-determination* are three new dimensions identified for consideration in terms of developing remote learning platforms targeted at S&T programs. These dimensions will reduce the shortcomings of existing frameworks. Due to limited time, several items have yet to be completed and are proposed for future work. The current findings will be complemented by a quantitative study. We thus recommend developing a prototype application based on the C19MOOC framework for a particular S&T course and conducting empirical studies with educators and students to gain deeper insight into the framework's feasibility. The findings obtained should be used to refine the framework. Although the C19MOOC framework is designed bearing S&T courses in mind, future studies should explore the use of this framework for other courses.

**Author Contributions:** Conceptualization, S.S. and J.S.D.; methodology, S.S.; software, S.S. and J.S.D.; validation, J.S.D., R.A. and M.A.F.; formal analysis, S.S.; investigation, S.S.; resources, S.S. and J.S.D.; data curation, S.S.; writing—original draft preparation, S.S.; writing—review and editing, S.S., J.S.D., R.A. and M.A.F.; visualization, S.S. and J.S.D.; supervision, J.S.D.; project administration, J.S.D.; funding acquisition, J.S.D. All authors have read and agreed to the published version of the manuscript.

**Funding:** This research was funded by UNITEN BOLD Refresh and RIPHEN JRP, grant code RIPHEN-JRP-PLU-MSU-06UNITEN.

**Institutional Review Board Statement:** Not applicable.

**Informed Consent Statement:** Not applicable.

**Data Availability Statement:** Not applicable.

**Acknowledgments:** We appreciate the time, patience, and insightful feedback provided by all participants in the FGD and expert review.

**Conflicts of Interest:** The authors declare no conflict of interest.

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
