# Peer review of "C19MOOC: A Remote Learning Reference Framework for Science and Technology Education"

_informatics, doi:10.3390/informatics9030053_

Round 1

Reviewer 1 Report

The Covid19 pandemic has turned remote learning from an option into a necessity. Unfortunately, the administratively introduced e-learning was treated as a necessary evil rather than as a new teaching system, creating new didactic opportunities.

It turned out that while technologically we are fully prepared to introduce remote learning on a massive scale, the methodology of such teaching practically does not exist. Previously appearing studies on this subject were addressed to a small group of specialists and enthusiasts.

The authors attempt to systematize the process of creating and evaluating e-learning projects. Such attempts are valuable and worth publishing.

The article specifies the factors that must be taken into account when creating remote learning projects. The list of factors and technologies selected appears to be complete.

The term M-learning introduced by the authors is misleading: it suggests preferential use of mobile technologies, whereas the procedures apply to any remote teaching.

What is missing from the paper:

Graphical structure of the M-learning system, flow diagrams, illustration of dependencies. These elements will increase the originality of the work and its usefulness.

The paper could end with a design tool, built, e.g., in the form of a checklist.

Author Response

Thank you very much for the feedback provided. We have made all the necessary amendments based on the feedback provided by all the reviewers. Please find attached our responses. We also used 'track changes' in the paper, so any amendments made can be easily seen.  

Reviewer 2 Report

The authors propose a useful framework for developing online learning, specifically m-learning, for Science and Technology Courses. Although the article presents some interesting review of the state of the art, it would be interesting to deepen the literature review regarding models and frameworks  not only to support the proposed framework (C19MOOC M-learning Framework) but also to bring greater richness to the discussion of the results of this research (section 5).

The research problem is not clear, so I suggest its enunciation at the end of the introduction. The methods are well described, however I didn't see the authors mentioning anything about the ethical approval they secured to use this data/information. This would need to be clarified in advance of any publication. Besides that I think it would be interesting to reflect upon the authors decision to involve such a large number of participants in the focus group. Add some points of view about the benefits and challenges of FG regarding this particular issue that you can find in the literature.

In the conclusions, the authors did not reflect on the limitations of the research, and did not pointed out some paths for future research. In the future, it is worth developing atheoretical and research model and conducting research on a large group in order to obtain results possible for statistical inference and practical application.

A careful revision of the text is advised to correct some errors and formatting problems.

Finally, there is merit in this manuscript in contributing to global discussion around online learning after an abruptly transition caused by the pandemic situation, specially in lab courses, we just need to ensure clarity.

Well done and I look forward to reading a revised draft.

Author Response

(The authors gave the same response as above.)

Reviewer 3 Report

1.  In the discussion part, there is only a brief description of the Focus Group Discussion, but failed to explore the reasons. It is suggested to conduct an in-depth analysis of the relationship between them, which can be combined with the interview part. Because the research results of the interview are not well reflected in the discussion.
2. It is suggested to deeply explain the reasons for Focus Group Discussion results. And there is no conclusion and future development direction.
3. what is the time and procedure of the interview, and how is the coding summary determined? Such content is not clearly explained in the article. 
4. There are still many errors in the overall writing format of the article and the English expressions. It is suggested that the author conducts English editing and refer to previous studies for reformation. 

Author Response

(The authors gave the same response as above.)

Round 2

Reviewer 1 Report

Only comments for the editors in this review

Author Response

We have amended the paper based on the feedback provided. Thank you. 

Reviewer 2 Report

Thank you for this resubmission. Many of my initial recommendations for the paper have been addressed and I am happy for this manuscript to continue through the process.

Author Response

(The authors gave the same response as above.)

Reviewer 3 Report

We have revisited the complete manuscript. Thank you very much again for your time and effort.

Author Response

(The authors gave the same response as above.)

Round 3

Reviewer 1 Report

In present form, the paper is acceptable for publication

This manuscript is a resubmission of an earlier submission. The following is a list of the peer review reports and author responses from that submission.